# Albumin-Bilirubin Score for Prediction of Outcomes in Heart Failure Patients Treated with Cardiac Resynchronization Therapy

**DOI:** 10.3390/jcm10225378

**Published:** 2021-11-18

**Authors:** Shinya Yamada, Takashi Kaneshiro, Akiomi Yoshihisa, Minoru Nodera, Kazuaki Amami, Takeshi Nehashi, Yasuchika Takeishi

**Affiliations:** 1Department of Cardiovascular Medicine, Fukushima Medical University, Fukushima 960-1295, Japan; tk2435@fmu.ac.jp (T.K.); yoshihis@fmu.ac.jp (A.Y.); nodera@fmu.ac.jp (M.N.); k-amami@fmu.ac.jp (K.A.); doryo9@fmu.ac.jp (T.N.); takeishi@fmu.ac.jp (Y.T.); 2Department of Arrhythmia and Cardiac Pacing, Fukushima Medical University, Fukushima 960-1295, Japan

**Keywords:** albumin-bilirubin score, cardiac resynchronization therapy, liver function, clinical outcomes

## Abstract

Background: Liver function may be a useful indicator of response to cardiac resynchronization therapy (CRT). We aimed to investigate the clinical significance of albumin-bilirubin (ALBI) score, an assessment tool of liver function, on outcomes in heart failure (HF) patients treated with CRT. Methods: We studied 180 patients undergoing CRT. The ALBI score, derived from albumin and total bilirubin, and left ventricular ejection fraction (LVEF) were assessed before and 6 months after CRT. The patients were classified according to the ALBI score before CRT; High (>−2.60) or Low (≤−2.60) ALBI groups. The patients were then reclassified based on the ALBI score before and 6 months after CRT; High/High, High/Low, Low/High, and Low/Low ALBI groups. We evaluated the prognostic value of the ALBI score for HF deaths after CRT. Results: During a median follow-up period of 50 months, there were 41 (22.7%) HF deaths. A Cox proportional hazard analysis revealed that high ALBI scores at baseline were not related to HF deaths (hazard ratio, 1.907, *p* = 0.068). However, High/High ALBI scores, but not High/Low or Low/High ALBI scores, were an independent predictor of HF deaths compared with Low/Low ALBI scores (hazard ratio, 3.449, *p* = 0.008), implying that consistently high ALBI scores were associated with poor prognosis. The percentage change in LVEF from baseline to 6 months after CRT did not differ among the four groups, suggesting that left ventricular systolic function was not linked with the ALBI score. Conclusions: ALBI scores before and after CRT are a new indicator of CRT response, and have a predictive value for HF deaths in HF patients.

## 1. Introduction

Cardiac resynchronization therapy (CRT) is regarded as a well-established therapy for patients with advanced heart failure (HF) [1]. Although there is a wide variability in the extent of ventricular remodeling and improvement in clinical conditions in patients treated with CRT, response to CRT is often determined based on the assessment of left ventricular (LV) systolic function after CRT [2,3]. To date, it has been recognized that HF is associated with multi-organ dysfunction, and cardio-hepatic syndrome is linked to poor prognosis and higher risk of death [4,5]. Therefore, liver function may also be a useful indicator of response to CRT. However, an adequate assessment tool of cardiogenic liver injury has not yet been fully established for HF patients treated with CRT.

The albumin-bilirubin (ALBI) score, which is calculated from the patient’s serum albumin and total bilirubin, has been proposed for assessing liver function and subsequent long-term mortality in patients with liver disease [6]. Generally, elevated bilirubin, a serum cholestasis marker, is related to acute liver congestion, and decreased albumin is associated with chronic congestive hepatopathy and poor prognosis in HF patients [7]. Therefore, the ALBI score might be suitable for assessing the severity of cardio-hepatic syndrome and subsequent prognosis. It has been demonstrated that high ALBI scores are associated with fluid overload and subsequent poor prognosis in HF patients [8]. Thus, we hypothesized that a response to CRT results in improved liver function and subsequent favorable outcomes in HF patients with a high ALBI score before CRT.

In the current study, we aimed to investigate the clinical significance of ALBI scores before and after CRT in the prognosis of HF patients.

## 2. Methods

### 2.1. Study Subjects

This was a retrospective study that enrolled 203 HF patients, who had undergone successful CRT device implantation between 2008 and 2020 at our institution. Before the CRT device implantation, a blood sampling test, 12-lead electrocardiography, and echocardiography, were performed in each patient to evaluate the clinical status. The eligibility criteria for CRT were New York Heart Association (NYHA) class II–IV symptoms of HF, despite receiving optimal medical therapy, LV ejection fraction (LVEF) of ≤35% on echocardiography, and a QRS duration of ≥120 ms, in accordance with the established criteria [9,10]. Additionally, we included patients who had indications of pacing for atrioventricular block; NYHA class II–III HF; and a LVEF of ≤50%, based on the Biventricular versus Right Ventricular Pacing in Heart Failure Patients with Atrioventricular Block (BLOCK HF) trial [11]. The choice of CRT with a defibrillator (CRT-D) or CRT with a pacemaker was based on the patient’s clinical history and arrhythmic risk profile. In the present study, nine patients who were positive for hepatitis B surface antigen and/or hepatitis C antibody, two patients who had chronic liver disease (cirrhosis and bile duct disease), and 12 patients who died within the 6-month follow-up period were excluded from this study. The remaining 180 patients constituted the final study population. Written informed consent was obtained from all study subjects, and the study protocol was approved by the Ethics Committee of Fukushima Medical University.

### 2.2. Device Implantation

Device implantation was performed when the HF was relatively compensated. First, the right atrial and ventricular leads were conventionally positioned in the right atrial appendage and the apex of the right ventricle, respectively. Next, a coronary sinus venogram was performed using a balloon catheter. The LV lead was inserted transvenously via the subclavian route. Then, the LV pacing lead was inserted through the coronary sinus and positioned in the venous system, preferably in a posterolateral vein and a non-apical site. Finally, all leads were connected to a biventricular cardiac device.

### 2.3. Data Collection

The laboratory and echocardiographic findings of the 180 patients implanted with CRT-D (*n* = 164) or CRT with a pacemaker (*n* = 16) were analyzed before the CRT device implantation. At baseline, the laboratory data included hemoglobin, platelets, albumin, total bilirubin, aspartate aminotransferase, alanine aminotransferase, lactate dehydrogenase, blood urea nitrogen, creatinine, estimated glomerular filtration rate, sodium, c-reactive protein, and brain natriuretic peptide. The measurement of estimated glomerular filtration rate was performed based on the Modification of Diet in Renal Disease formula [12]. Echocardiographic parameters included left atrial diameter, LV end-diastolic diameter, LV end-systolic diameter, LV end-diastolic volume index, LV end-systolic volume index, LVEF, tricuspid regurgitation pressure gradient, and inferior vena cava diameter.

### 2.4. Determination of Risk Factors

Target comorbidities, such as hypertension, diabetes and chronic kidney disease were determined as described in a previous study [13]. Hypertension was defined as the recent use of antihypertensive drugs, systolic blood pressure ≥140 mmHg, and/or diastolic blood pressure >90 mmHg. Diabetes was defined as the recent use of insulin or antidiabetic drugs, a fasting blood glucose value of >126 mg/dL, and/or a hemoglobin A_1_c value of >6.5%. Chronic kidney disease was defined as an estimated glomerular filtration rate of <60 mL/min/1.73 m^2^ [12].

### 2.5. Assessment of ALBI Score before CRT

The ALBI score was calculated as follows: (log10 total bilirubin [mmol/L] × 0.66) + (albumin [g/L] × −0.085) [6]. In a study by Johnson et al. [6], the ALBI score was developed as a reliable maker for the assessment of liver function, and an ALBI score of ≤−2.60 was defined as a good prognostic indicator for survival in patients with liver disease. We then divided the patients (*n* = 180) into two groups based on their ALBI score before CRT: the High ALBI group (>−2.60, *n* = 109) and the Low ALBI group (≤−2.60, *n* = 71). The laboratory findings, echocardiographic data, and outcomes were compared between the two groups.

### 2.6. Reassessment of ALBI Score 6 Months after CRT

We also investigated the relationship between changes in the ALBI score 6 months after CRT and outcomes. The patients were reclassified into the following four groups based on their ALBI scores before and 6 months after CRT: the High/High group (ALBI score > −2.60 both before and after CRT); the High/Low group (ALBI score > −2.60 before CRT and ≤ −2.60 after CRT); the Low/High group (ALBI score ≤ −2.60 before CRT and > −2.60 after CRT); and the Low/Low group (ALBI score ≤ −2.60 both before and after CRT). We compared the percentage change in LVEF from baseline to 6 months after CRT as an assessment of LV systolic function, as well as outcomes, among the four groups.

### 2.7. Identification of Adverse Events during the Follow-Up

The follow-up of adverse events was continued until March 2021. The clinical endpoints of this study were an HF death and a lethal arrhythmic event. A HF death was confirmed by independent attending physicians using an HF guideline [14]. A lethal arrhythmic event was defined as an appropriate implantable cardioverter-defibrillator (ICD) therapy (anti-tachycardia pacing or shock therapy) or sudden cardiac death. Programming of the ICD therapy was performed at the cardiologist’s discretion according to the patient’s background. The definition of sudden cardiac death was the unexpected death of an individual not attributable to an extracardiac cause, within 1 h of symptom onset [15]. The status and/or dates of death of all patients were obtained from the patients’ medical records or attending physicians at the patient’s referring hospital. All patients received follow-up in this study. The survival time was calculated from the date of CRT device implantation until the date of adverse events or last follow-up.

### 2.8. Statistical Analysis

Normally distributed data are reported as mean ± standard deviation, and non-normally distributed data are reported as median and interquartile range. Categorical variables were expressed as numbers and percentages. In the normally distributed data, an independent sample *t* test was used to analyze the differences between the High and Low ALBI groups. In the non-normally distributed data, differences between the two groups were compared using a Mann–Whitney U test. Categorical data were analyzed using a chi-square test. Differences among the four groups before and after CRT were assessed utilizing an analysis of variance for comparison of parametric continuous variables and the Kruskal–Wallis test for comparison of non-parametric continuous variables. The Cox proportional hazard regression analysis was performed to clarify the relationship between ALBI score and the incidence of adverse events. To prepare for potential confounding, we considered the following clinical factors, which are generally known to affect prognosis in HF patients treated with CRT: left bundle branch block morphology, NYHA class III/IV, brain natriuretic peptide and LVEF. Those variables were selected for testing in a multivariable analysis. The cumulative incidence curve of adverse events was plotted via the Kaplan–Meier method, with statistical significance examined using the log-rank test. A value of *p* < 0.05 was considered statistically significant. Statistical analyses were performed with the SPSS statistical software (version 27.0, SPSS Institute, Chicago, IL, USA).

## 3. Results

### 3.1. Baseline Characteristics

The baseline clinical characteristics of the present study’s subjects are summarized in Table 1. The study subjects were divided into the High ALBI group (*n* = 109) and Low ALBI group (*n* = 71) based on their ALBI score before CRT [6]. At baseline, the ratios of NYHA class III/IV and use of inotropic agents were significantly higher in the High ALBI group than in the Low ALBI group. Systolic blood pressure was significantly lower in the High ALBI group than in the Low ALBI group. However, diastolic blood pressure and heart rate were similar between the two groups. In addition, there were no significant differences between the two groups in any other data, as shown in Table 1.

### 3.2. Laboratory and Echocardiographic Data

In the laboratory data (Table 2), levels of hemoglobin and albumin were significantly lower, whereas levels of total bilirubin, c-reactive protein and brain natriuretic peptide, as well as ALBI score were significantly higher in the High ALBI group than in the Low ALBI group. However, there were no significant differences between the two groups in any other data, as shown in Table 2.

In the echocardiographic data, the tricuspid regurgitation pressure gradient and inferior vena cava diameter were significantly higher in the High ALBI group than in the Low ALBI group. However, the left atrial diameter, LV end-diastolic diameter, LV end-systolic diameter, LV end-diastolic volume index, LV end-systolic volume index, and LVEF did not differ between the two groups.

### 3.3. ALBI Score at Baseline and Adverse Events

During a median follow-up period of 50 months, there were 41 (22.7%) HF deaths. The incidence of HF death in the High ALBI group was 27.5%, and that in the Low ALBI group was 15.4%. In the Kaplan–Meier analysis, the risk of HF deaths was comparable between the High and Low ALBI groups (log-rank *p* = 0.063) as shown in Figure 1A. In the Cox proportional hazards model, a high ALBI score was not associated with HF deaths with a hazard ratio (HR) of 1.907 (95% confidence interval: 0.954–3.813, *p* = 0.068).

We also investigated the relationship between the ALBI score and lethal arrhythmic events. During a median follow-up period of 31 months, there were 79 (43.8%) lethal arrhythmic events (73 appropriate ICD therapies and six sudden cardiac deaths). The incidence of lethal arrhythmic events was 43.1% in the High ALBI group, and that in the Low ALBI group was 45.0%. In the Kaplan–Meier analysis, a risk of lethal arrhythmic events was comparable between the High and Low ALBI groups (log-rank *p* = 0.725) as shown in Figure 1B. The Cox proportional hazards model revealed that a high ALBI score was not associated with lethal arrhythmic events, with an HR of 1.084 (95% confidence interval: 0.689–1.704, *p* = 0.727).

In the receiver-operating characteristic analysis, the optimal cut-off value of the ALBI score at baseline for predicting HF death was −2.42 (95% confidence interval: 0.524–0.722, *p* = 0.015, sensitivity of 63.4%, specificity of 58.3%, and areas under the curve of 0.62), and the optimal cut-off value of ALBI score at baseline for predicting lethal arrhythmic event was −2.46 (95% confidence interval: 0.415–0.585, *p* = 0.999, sensitivity of 50.6%, specificity of 49.5%, and areas under the curve of 0.50).

### 3.4. Changes in the ALBI Score after CRT and Adverse Events

In the present study, 109 (60.5%) and 84 (46.6%) of the patients had high ALBI scores before and after CRT, respectively. The patients (*n* = 180) were recategorized into four groups based on their ALBI scores before and 6 months after CRT: High/High (*n* = 62), High/Low (*n* = 47), Low/High (*n* = 22) and Low/Low (*n* = 49). The percentage change in LVEF from baseline to 6 months after CRT was calculated as an assessment of LV systolic function, and it was not different among the four groups (Figure 2).

We next investigated the relationship between changes in the ALBI score after CRT and HF deaths. The incidence of HF deaths was observed in 20 (32.2%) of the High/High ALBI group, 10 (21.2%) of the High/Low ALBI group, 5 (22.7%) of the Low/High ALBI group, and 6 (12.2%) of the Low/Low ALBI group. As seen in Figure 3A, the Kaplan–Meier analysis showed that the High/High ALBI group had the highest rate of HF death (log-rank *p* = 0.033). The Cox proportional hazard analysis showed that the High/High ALBI scores were an independent predictor for HF deaths compared with the Low/Low ALBI scores (HR, 3.449, *p* = 0.008), as shown in Table 3. However, no significant differences were observed in the High/Low and Low/High ALBI scores regarding the incidence of HF deaths compared with the Low/Low ALBI scores. In a multivariable model after adjusting for the left bundle branch block morphology, NYHA class III/IV, brain natriuretic peptide and LVEF, High/High ALBI scores were significantly associated with HF deaths (HR, 2.687, *p* = 0.040).

We also investigated the relationship between changes in the ALBI score after CRT and lethal arrhythmic events. Lethal arrhythmic events occurred in 29 patients (46.7%) of the High/High ALBI group, 18 (38.2%) of the High/Low ALBI group, 10 (45.4%) of the Low/High ALBI group, and 22 (44.8%) of the Low/Low ALBI group. As seen in Figure 3B, the Kaplan–Meier analysis showed that the risk of lethal arrhythmia was not different among the four groups (log-rank *p* = 0.582). The Cox proportional hazard analysis showed that no significant differences were observed in the four groups regarding lethal arrhythmic events (Table 3).

Furthermore, we investigated the relationship between the ALBI score at 6 months after CRT and outcomes. The Cox proportional hazard analysis revealed that high ALBI scores at 6 months after CRT were related to HF deaths (HR, 2.346, *p* = 0.008). However, high ALBI scores at 6 months after CRT were not associated with lethal arrhythmic events (HR, 1.346, *p* = 0.188).

## 4. Discussion

In the present study, we investigated whether the ALBI score, an assessment tool of liver function, can predict HF deaths in patients treated with CRT. The results showed that the patients with a high ALBI score of >−2.60 had a higher risk of HF deaths before CRT (lower hemoglobin, higher c-reactive protein and brain natriuretic peptide), but the event rate of HF deaths was comparable between the patients with high and low ALBI scores. Importantly, high ALBI scores both before and after CRT were associated with a high risk of HF deaths, whereas a decrease in ALBI scores to ≤−2.60 after CRT was related to a low risk of HF deaths. These findings indicate that the assessment of ALBI score before and after CRT enables us to predict HF deaths.

### 4.1. Assessment of ALBI Score in HF Patients before CRT

Generally, two distinct forms are recognized as cardiogenic liver injury in patients with HF: (1) passive congestion, which is caused by an increase in central venous pressure; and (2) impaired perfusion, which is caused by a decrease in cardiac output [16]. Although patients with cardiogenic liver injury are often asymptomatic, a blood examination is useful for the diagnosis of the two distinct liver injuries. Passive congestion is mainly associated with elevation of cholestatic enzymes, and impaired perfusion is mainly related to elevation of transaminase [17,18]. In the present study, we used the ALBI score as an assessment tool of liver function for HF patients treated with CRT. The ALBI score, derived from a patient’s albumin and total bilirubin, was originally developed in patients with hepatocellular carcinoma for the reliable assessment of liver function, and an ALBI score of >−2.60 was proposed as a prognostic indicator for mortality [6]. It is well-known that levels of total bilirubin, a serum cholestasis marker, and albumin are linked with central venous pressure in patients with HF [4,18]. Thus, it can be postulated that the severity of liver congestion in HF patients is characterized by their ALBI score. In the current study, patients with a high ALBI score had a higher tricuspid regurgitation pressure gradient and inferior vena cava diameter compared with those with a low ALBI score on echocardiography, suggesting that a high ALBI score is associated with increased central venous pressure. Furthermore, patients with a high ALBI score had a higher c-reactive protein and brain natriuretic peptide, as well as lower hemoglobin, compared with those with a low ALBI score. These cardiovascular risk factors are highly prevalent in patients with HF, and they generally contribute to the incidence of HF deaths [19,20,21,22]. However, the event rate of HF deaths was comparable between the patients with high and low ALBI scores during the follow-up period. These contrasting results might be explained by an improvement in liver function achieved by CRT, indicating that reassessment of the ALBI score after CRT may be required to predict prognosis accurately.

### 4.2. Reassessment of ALBI Score in HF Patients after CRT

It has been revealed that intraventricular, atrioventricular, and interventricular synchronies can be improved by CRT [23]. These beneficial effects of CRT may result in improved liver function and subsequent favorable outcomes. It has been estimated that as many as 15–65% of patients with advanced HF have congestive hepatopathy [4,5]. In the present study, 109 (60.5%) of the patients had high ALBI scores before CRT, indicating that they had cardiogenic liver injury. Importantly, high ALBI scores both before and after CRT were related to a high risk of HF deaths, but a decrease in the ALBI score from >−2.60 to ≤−2.60 after CRT was associated with a low risk of HF deaths. These results suggest that an improvement in liver congestion leads to better clinical outcomes, as a result of CRT. Although the response to CRT is often assessed based on LV systolic function [2,3,24], the percentage change in LVEF from baseline to 6 months after CRT did not differ among the patients with different ALBI scores in the present study, suggesting that LV systolic function is not linked with the ALBI score. Therefore, the ALBI score is a new indicator of CRT response (presumably improved interventricular synchrony) and subsequent favorable outcomes.

The relationship between liver function and lethal arrhythmic events is not yet fully understood in HF patients treated with CRT. In the current study, changes in the ALBI score were not associated with the incidence of lethal arrhythmic events after CRT.

### 4.3. Clinical Implication

The BLOCK HF trial reported the beneficial effects of CRT on patients who had indications of pacing for atrioventricular block; NYHA class I–III HF; and a LVEF of ≤50% [11]. However, most clinical CRT trials exclude mild to moderate HF patients with atrioventricular block, and do not provide adequate information on the clinical management of HF patients treated with CRT. Therefore, in the present study, we included mild to moderate HF patients with atrioventricular block, based on the BLOCK HF classification. Consequently, we clearly demonstrated for the first time that the assessment of the ALBI score provides prognostic information on HF deaths after CRT. From our study results, reassessment of the ALBI score after CRT is required for the accurate prediction of HF deaths. A high ALBI score of >−2.60 both before and after CRT is associated with a poor prognosis. An improvement in the ALBI score from >−2.60 to ≤−2.60 after CRT indicates better clinical outcomes.

### 4.4. Limitations

There are some limitations to the current study. First, it was performed in a single institution with a relatively small number of subjects. Second, the right ventricular function was not fully assessed to confirm interventricular dyssynchrony. Third, pharmacological neurohormonal and hemodynamic control may reduce the risk of death. In the present study, the usage of β blockers and renin-angiotensin-aldosterone system inhibitors was similar between the High and Low ALBI groups. However, the relationship between doses of the used drugs and outcomes could not be statistically analyzed in the present study, because many kinds of drugs (β blockers and renin-angiotensin-aldosterone system inhibitors) were used as optimal therapy for HF. Finally, we performed a multivariable Cox proportional hazard analysis to validate the relationship between the ALBI score and adverse events. However, interactions between the ALBI score and adverse events during the follow-up period could not be fully adjusted. We are considering conducting a further study to address these issues.

## 5. Conclusions

Our results indicate that the ALBI score is a new indicator of CRT response, and reassessment of the score after CRT is required for the accurate prediction of HF deaths. A high ALBI score of >−2.60 both before and after CRT is associated with poor outcomes, whereas a decrease in the ALBI score from >−2.60 to ≤−2.60 after CRT indicates favorable outcomes.

## Figures and Tables

**Figure 1 jcm-10-05378-f001:**
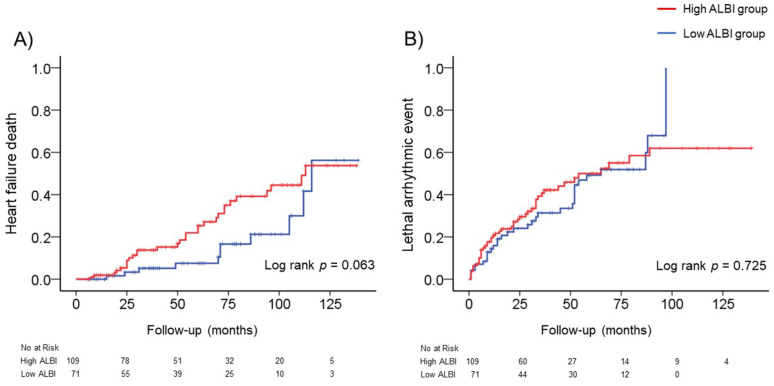
The incidence of adverse events (heart failure death or lethal arrhythmic event) in the High and Low ALBI groups. (**A**) The cumulative incidence curve of heart failure death with a log rank test between the High and Low ALBI groups. (**B**) The cumulative incidence curve of lethal arrhythmic events with a log rank test between the High and Low ALBI groups.

**Figure 2 jcm-10-05378-f002:**
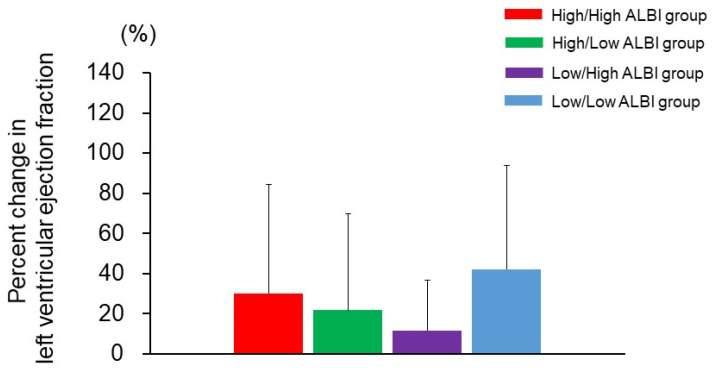
The percentage change in left ventricular ejection fraction from baseline to 6 months after cardiac resynchronization therapy among the four groups.

**Figure 3 jcm-10-05378-f003:**
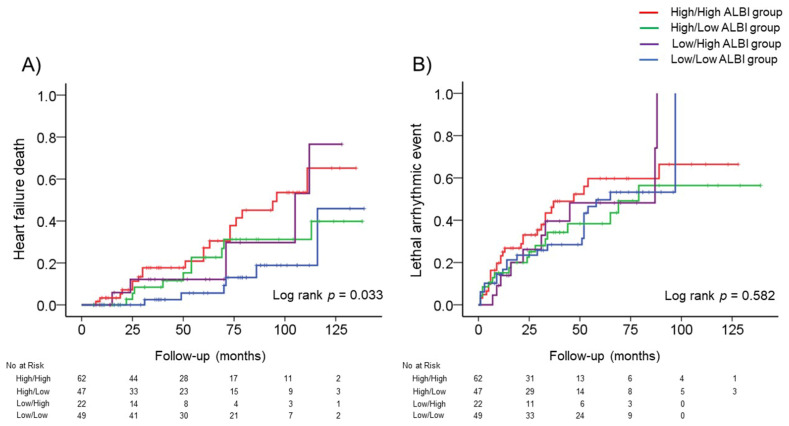
The incidence of adverse events (heart failure death or lethal arrhythmic event) among the four groups. (**A**) The cumulative incidence curve of heart failure death with a log rank test among the four groups. (**B**) The cumulative incidence curve of lethal arrhythmic events with a log rank test among the four groups.

**Table 1 jcm-10-05378-t001:** Comparison of baseline clinical characteristics for the High and Low ALBI groups.

	High ALBI Group (*n* = 109)	Low ALBI Group (*n* = 71)	*p* Value
Age	67.2 ± 11.5	66.7 ± 10.9	0.770
Male (*n*, %)	77 (70.6%)	50 (70.4%)	0.975
Body mass index, kg/cm^2^	22.2 ± 3.6	22.5 ± 3.6	0.529
Systolic blood pressure, mmHg	102.0 (94.0–112.0)	109.0 (96.0–118.0)	0.030
Diastolic blood pressure, mmHg	60.0 (56.0–68.0)	63.0 (59.0–70.0)	0.068
Heart rate, bpm	65.9 ± 15.4	63.3 ± 14.9	0.266
NYHA class III/IV (*n*, %)	46 (42.2%)	17 (23.9%)	0.012
Ischemic etiology (*n*, %)	26 (23.8%)	10 (14.0%)	0.109
Hypertension (*n*, %)	68 (62.3%)	41 (57.7%)	0.534
Diabetes (*n*, %)	59 (54.1%)	32 (45.0%)	0.235
Chronic kidney disease (*n*, %)	76 (69.7%)	52 (73.2%)	0.611
Atrial fibrillation (*n*, %)	30 (27.5%)	13 (18.3%)	0.157
QRS duration >150 ms (*n*, %)	70 (64.2%)	48 (67.6%)	0.640
LBBB morphology (*n*, %)	60 (55.0%)	44 (61.9%)	0.358
Implantation for secondary prevention (*n*, %)	39 (35.7%)	24 (33.8%)	0.786
Atrioventricular block requiring a pacemaker (*n*, %)	24 (22.4%)	12 (16.9%)	0.369
* **Medication** *			
β blockers (*n*, %)	109 (100.0%)	68 (95.7%)	0.060
ACE-Inhibitors/ARBs (*n*, %)	91 (83.4%)	59 (83.0%)	0.946
Mineralocorticoid receptor antagonists (*n*, %)	69 (63.3%)	52 (73.2%)	0.165
Amiodarone (*n*, %)	66 (60.5%)	36 (50.7%)	0.193
Inotropic agents (*n*, %)	51 (46.7%)	16 (22.5%)	0.001

ACE-inhibitors: angiotensin-converting enzyme-inhibitors; ARBs: angiotensin II receptor blockers; NYHA: New York Heart Association; LBBB: left bundle branch block.

**Table 2 jcm-10-05378-t002:** Comparison of baseline laboratory and echocardiographic data for the High and Low ALBI groups.

	High ALBI Group (*n* = 109)	Low ALBI Group (*n* = 71)	*p* Value
*Laboratory data*			
Hemoglobin (g/dL)	12.4 ± 2.2	13.4 ± 1.7	0.002
Platelet (×10^3^/μL)	172.0 (147.5–215.5)	168.0 (150.0–211.0)	0.701
Albumin (g/dL)	3.4 ± 0.4	4.2 ± 0.2	<0.001
Total bilirubin (mg/dL)	1.0 ± 0.7	0.8 ± 0.3	0.036
Aspartate aminotransferase (U/L)	25.0 (19.5–32.0)	26.0 (20.0–35.0)	0.459
Alanine aminotransferase (U/L)	21.0 (13.5–32.5)	22.0 (13.0–33.0)	0.645
Lactate dehydrogenase (U/L)	199.0 (167.0–251.0)	213.5 (185.5–240.0)	0.129
Blood urea nitrogen (mg/dL)	20.5 (16.2–27.0)	21.0 (17.0–29.0)	0.519
Creatinine (mg/dL)	1.09 (0.83–1.41)	1.09 (0.92–1.38)	0.816
eGFR (mL/min/1.73 m^2^)	48.0 (37.0–68.0)	52.0 (37.0–61.0)	0.965
Sodium (mEq/L)	138.0 (136.0–140.0)	139.0 (137.0–141.0)	0.320
C-reactive protein (mg/dL)	0.24 (0.08–0.72)	0.11 (0.05–0.21)	<0.001
Brain natriuretic peptide (pg/mL)	379.7 (184.8–615.3)	190.5 (121.8–340.0)	<0.001
ALBI score	−2.15 ± 0.32	−2.89 ± 0.22	<0.001
*Echocardiographic data*			
Left atrial diameter (mm)	46.1 ± 8.7	45.9 ± 9.7	0.870
LV end-diastolic diameter (mm)	62.8 ± 9.4	61.7 ± 8.7	0.422
LV end-systolic diameter (mm)	54.5 ± 11.2	52.9 ± 10.0	0.335
LV end-diastolic volume index (mL/m^2^)	110.2 ± 47.1	109.8 ± 46.6	0.956
LV end-systolic volume index (mL/m^2^)	79.4 ± 41.5	80.0 ± 38.0	0.921
LV ejection fraction (%)	30.2 ± 11.0	30.9 ± 9.3	0.630
TRPG (mmHg)	28.3 ± 11.4	24.7 ± 8.4	0.034
Inferior vena cava diameter (mm)	15.8 ± 5.3	14.2 ± 4.4	0.041

eGFR: estimated glomerular filtration rate; LV: left ventricular; TRPG: tricuspid regurgitation pressure gradient.

**Table 3 jcm-10-05378-t003:** Cox proportional hazard model of heart failure deaths and lethal arrhythmic events.

**Heart Failure Deaths** **(41 Events/180 Patients)**	**Hazard Ratio**	**95% Confidence Interval**	***p* Value**
Low/Low ALBI group	Reference	–	–
Low/High ALBI group	2.765	0.842–9.086	0.094
High/Low ALBI group	1.852	0.669–5.129	0.235
High/High ALBI group	3.449	1.383–8.600	0.008
High/High ALBI group *	2.687	1.047–6.899	0.040
**Lethal Arrhythmic Events** **(79 Events/180 Patients)**	**Hazard Ratio**	**95% Confidence Interval**	* **p** * **Value**
Low/Low ALBI group	Reference	–	–
Low/High ALBI group	1.189	0.562–2.515	0.651
High/Low ALBI group	0.922	0.493–1.724	0.799
High/High ALBI group	1.339	0.766–2.340	0.306

* Adjusted: adjusted for left bundle branch block morphology, New York Heart Association class III or IV, brain natriuretic peptide and left ventricular ejection fraction.

## Data Availability

The findings of the present study are available on request from the corresponding author. The data are not publicly available due to privacy or ethical restrictions.

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
