# Peer review of "Albumin-Bilirubin Score for Prediction of Outcomes in Heart Failure Patients Treated with Cardiac Resynchronization Therapy"

_jcm, 2021, doi:10.3390/jcm10225378_

Round 1

Reviewer 1 Report

The manuscript ,, Albumin-bilirubin score for prediction of outcomes in heart failure patients treated with cardiac resynchronization therapy’’ address an important and unsolved problem of assessing the effectiveness of therapy and prognosis in patients with HF using simple indicators. In the present study, the authors investigated whether the ALBI score, an assessment tool of liver function, can predict HF deaths in patients treated with CRT.  This study showed that high ALBI scores both before and after CRT were associated with a high risk of HF deaths, whereas a decrease in ALBI scores to ≤ -2.60 after CRT was related to a low risk of HF deaths. These findings indicate that the assessment of ALBI score before and after CRT enables us to predict HF deaths. The inclusion and exclusion criteria are correctly defined. The study has been well planned, the methodology and statistics are adequate. However, I have a following minor comments:

  1. What percentage of patients underwent OHT and LVAD implantation during the follow-up? Were these patients excluded from the study?
  2. Optimal therapy for heart failure should consist of a beta-blocker, ACEI / ARB, and MRA. What percentage of patients was treated with MRA? Were there differences between the analyzed groups in the use of MRA? Were the patients treated with ARNI in the analyzed group?
  3. For better understanding the study results it is reasonable to present doses of the used drugs and they relation with outcome . Complete pharmacological neurohormonal and hemodynamic control may reduce the risk of death , rehospitalisation. I strongly encourage the Authors to present it.
  4. HR and blood pressure should be added to patients characteristic - especially mean HR is crucial for better understanding patients clinical status.
  5. Lines 154-160 and 165-171 repeat the results from Tables I and II. Please delete this phrases and comment briefly the results presented in Tables I and II.
  6. Is it possible to present the prognostic value of the ALBI using the ROC curve?

Reviewer 2 Report

I congratulate the authors for the paper and would suggest some clarifications.

In the baseline characteristics there are 36 patients with atrioventricular block requiring a pacemaker (
based on the BLOCK HF trial?) while only 16 patients were implanted with CRTP. The authors should explain this discrepancy.

Although deducible from the data, I would suggest an analysis of how many patients were in high and low ALBI scores before and after 6 months of CRT, which would indicate the percentage of patients improved in terms of liver function after CRT.

Consequently an analysis would be appreciable of ALBI score at 6 month after CRT and predictivity of adverse events.

Correct "sores" with "scores" at row 270
